# High-Pressure-Resistant Flexible Seven-in-One Microsensor Embedded in High-Pressure Proton Exchange Membrane Water Electrolyzer for Real-Time Microscopic Measurement

**DOI:** 10.3390/membranes12100919

**Published:** 2022-09-22

**Authors:** Chi-Yuan Lee, Chia-Hung Chen, Shan-Yu Chen, Hsiao-Te Hsieh

**Affiliations:** 1Department of Mechanical Engineering, Yuan Ze Fuel Cell Center, Yuan Ze University, Taoyuan 32003, Taiwan; 2Homytech Global Co., Ltd., Taoyuan 33464, Taiwan

**Keywords:** PEMWE, MEMS, flexible seven-in-one microsensor, hydrogen production, real-time monitor, aging, high pressure

## Abstract

The high-pressure proton exchange membrane water electrolyzer (PEMWE) used for hydrogen production requires a high-operating voltage, which easily accelerates the decomposition of hydrogen molecules, resulting in the aging or failure of the high-pressure PEMWE. As the high-pressure PEMWE ages internally, uneven flow distribution can lead to large temperature differences, reduced current density, flow plate corrosion, and carbon paper cracking. In this study, a new type of micro hydrogen sensor is developed with integrated flexible seven-in-one (voltage; current; temperature; humidity; flow; pressure; and hydrogen) microsensors.

## 1. Introduction

The technology of the hydrogen energy and fuel cell industry is becoming more sophisticated worldwide. As such, there is the expectation of accelerating the application of hydrogen energy to daily life and thus achieving the effect of energy saving and carbon reduction. The Hydrogen Council authorized McKinsey & Company to publish a report on the vision for hydrogen energy [1], which indicates that hydrogen energy will represent 18% of energy use in 2050. The report on the differences in annual carbon emissions—published by the United Nations Environment Programme [2]—indicates that global carbon emissions must be reduced by 7.6% annually, until 2030, to limit the temperature rise to within the ideal target of 1.5°C, as per the Paris Agreement. According to the hydrogen fuel cell market analysis report, published by the international survey institution of Market Research Future in 2021 [3], the market scale of fuel cells in relation to hydrogen energy is estimated to be USD 2 billion by 2027. Further, the annual growth rate will be 31.4% in the following seven years, indicating that this growth space has a high potential. Therefore, to reduce the reliance on fossil fuels, governments will have to focus on renewable energy [4]. However, due to the intermittency, in order to solve the peak issue, the realization of peak shaving becomes the primary issue. However, there is a severe defect in the present renewable energy, i.e., ‘seasonal influence’. Even on a daily basis, solar energy has peaks and valleys during energy generation. This power cannot be supplied continuously and stably, as such existing renewable energy cannot be extensively used in the context of people’s livelihoods and industry. Moreover, electrical energy storage technology is the key factor in whether renewable energy can reach an economic scale [5,6]. In terms of energy storage and intelligent system integration, energy storage equipment is arranged to allow renewable energy to dispatch and stabilize the electricity generation of renewable energy and demand, and therefore to provide system resilience and a spinning reserve [7]. 

Chen et al. [8] mentioned that the voltage uniformity of cells in the cell stack was the key factor in their service life. Liu et al. [9] stated that the cell stack depended on the performance of the weakest single cell; therefore, increasing the voltage uniformity and preventing performance degradation of the cell stack are very important. Shin et al. [10] found that the combination of PEMWE and PEMFC can act as an energy storage system. Ogumerem et al. [11] indicated that high working temperatures could improve the performance and power supply efficiency of the PEMWE system; this is because the required minimum energy decreased as the temperature rose. However, the stability of the membrane became worse when the temperature was higher than 100 °C, as such the operating temperature range should be limited to between 70~90 °C, which makes it very important to instantly monitor and control the optimum operating temperature of PEMWE. Li et al. [12] indicated that when the PEMWE was producing hydrogen, the activity of water molecules in the air was higher, and the system performance also decayed faster under a high applied potential. Therefore, aging was faster when the applied voltage and humidity were high. Yin et al. [13] found that the water absorption of PEMWE increased significantly when the absolute humidity was about 30 gm^−3^. In the 85 °C working condition, the variation in relative humidity (RH) induced higher absolute humidity, leading to higher water absorption of the catalyst layer. Zhao et al. [14] found that the saturation pressure of water vapor increased with temperature, and that the RH inside the runner decreased accordingly. Afshari et al. [15] found that all of the losses of PEMWE were related to the water transmission mechanism, which itself resulted from the electro-osmotic drag, pressure difference, and diffusion between the anode and cathode sides. In addition, the membrane thickness, cathode pressure, and operating temperature all influenced the exchange of hydrogen. The simulation and test results of Liu et al. [16] proved that after the PEMWE had higher contact pressure, it had better electrochemical properties, and the PEMWE with a pneumatic clamping mechanism had a better distribution of the contact pressure. Cai et al. [17] studied the synthetic Pd nanoparticle-modified SnO_2_ nanowire and manufactured a high-sensitivity and high-selectivity hydrogen sensor. The response was 55.72, when 60Pd- SnO_2_ was exposed to 300 °C and 100 ppm hydrogen, which is 12.7 times the response of bare SnO_2_. Ambardekar et al. [18] mentioned that the SnO_2_ sensor showed resistance stability, continuous response, and repeatability after the circulation was restored and when the gas response was not influenced. 

In compliance with the policy trend and internal real-time monitoring of the high-pressure PEMWE, this study used the micro-electro-mechanical systems (MEMS) technology to integrate the micro voltage, current, temperature, humidity, flow, pressure, and micro hydrogen sensors on a high-temperature-resistant, corrosion-resistant, stretchable, and flexible PI substrate. This is after optimization design, process optimization, and the application of prior experience. Lee et al. [19] showed that flexible six-in-one (micro temperature, humidity, flow, pressure, voltage, and current sensors) microsensors were successfully integrated onto a 50 μm thick Polyimide (PI) substrate by using micro-electro-mechanical systems (MEMS) technology. After the optimal design and process optimization of the flexible six-in-one microsensor was achieved, it was embedded into the PEMWE for a 500 h persistent effect test and internal real-time microscopic monitoring. Thus, as with the integrated flexible six-in-one microsensor, in this study, a flexible seven-in-one microsensor was developed and embedded in the high-pressure PEMWE for internal real-time microscopic monitoring.

## 2. Process Development of Flexible Seven-in-One Microsensor

In this study, MEMS were used to develop a flexible seven-in-one microsensor which can be embedded in a high-pressure PEMWE for real-time monitoring. Figure 1 shows the design drawing of the flexible seven-in-one microsensor, and the mask layout of the flexible seven-in-one microsensor is shown in Figure 2. Appropriate process materials were used, so that accurate, internal real-time microscopic monitoring information could be obtained in the electrochemical environment of a high-pressure water electrolyzer. 

In terms of process, surface micromachining technology was used, including lithography, metal deposition, and metal lift off. The fabrication process is shown in Figure 3. 

(a)PI film cleaning and fixing

The sample should be soaked and cleaned in acetone and methanol, in turn, before deposition, and cleaned by an ultrasonic oscillator for three minutes. 

(b)Circuit layer lithography 

The sample was uniformly coated with the spin-coater positive photoresist (AZ^®^ P4620) and exposed by a double-side aligner. The development was performed after exposure. The developer used for this experiment was AZ^®^ 400K, which was chosen to avoid too fast of a development—as this would result in overdevelopment of the defined pattern—as well as too thin of a line, and a lower yield. In this experiment, the developer and DI water were mixed in a ratio of 1:4 for the purposes of development, the developing time was about three minutes, and then the complete pattern was obtained. 

(c)Metal deposition 

The metal was deposited via an e-beam evaporator. The 300Å thick Cr and 1500Å Au were deposited at a deposition rate of 0.5-1Å/s. 

(d)Metal lift off 

The lift off was performed after deposition. The original photoresist was removed by using acetone in the lift-off process, while the excess metal was lifted off, and only the metal of the electrode pattern was left on the sample. 

(e)Protection layer 

Fujifilm Durimide^®^ PI 9320 (Fujifilm, Tokyo, Japan) was used as the material for the protection layer. The purpose of the secondary exposure and development is to complete the insulation protection layer with a high mechanical strength that is adapted to a highly chemical environment. 

(f)Dielectric layer of the humidity sensor 

Fujifilm Durimide^®^ PI 9305 was used as the humidity-sensitive, thin-film, micro humidity sensor in this experiment. 

(g)Dielectric layer of the pressure sensor 

The dielectric layer material of the pressure sensor was stretchable Fujifilm Electronic Materials U.S.A., Inc. LTC^®^ 9305 (North Kingstown, RI, USA), which has high mechanical strength, corrosion resistance, and resistance to an electrochemical environment. 

(h)Upper circuit of the pressure sensor 

The non-covered area was protected by AZ^®^ P4620. Moreover, the Au was coated by the e-beam evaporator on the substrate, and the original photoresist (which was removed by acetone lift off) was completed last. 

(i)Fabrication of hydrogen sensor

The non-covered area with SnO_2_ was protected by AZ^®^ P4620, and SnO_2_ and Pt were deposited on the substrate using the e-beam evaporator. The photoresist that was coated on the substrate was removed last, and this was achieved by using acetone to achieve the lift-off effect. 

To avoid the high-pressure-resistant, flexible seven-in-one microsensor being damaged by the closing pressure of the end plate in the high-pressure PEMWE, the insulation protection layer must have high strength. 

The purpose of the protective layer is to bring the sensing area of the miniature voltage and current sensors into direct contact with the flow channel ribs and to provide signal transmission and output. The optical micrograph of the completed, flexible seven-in-one microsensor is shown in Figure 4.

## 3. Correction of the Flexible Seven-in-One Microsensor

The flexible seven-in-one microsensor should be corrected, after fabrication, to measure signals and to verify the reliability. As such, the flexible seven-in-one micro-sensors were corrected one by one; the microsensor was corrected three times and the average value was taken to guarantee accuracy. However, this cannot be embedded in the high-pressure PEMWE for internal real-time microscopic monitoring until its reliability is confirmed.

### 3.1. Correction of Micro Temperature Sensor 

The environment for microsensor correction simulated the true environment conditions; this study used a program-controlled constant temperature and humidity testing machine (Hung ta HT-8045A Environmental Chamber) as the basis for the correction environment. The runner was full of DI water when the high-pressure PEMWE was in operation; therefore, the humidity was fixed at 100% during the course of temperature correction. The temperature correction range of the three micro temperature sensors was 20 °C to 90 °C. One signal was captured at an interval of 10 °C, beginning at 20 °C, and eight signals were captured by the NI PXI data acquisition unit and were nondimensionalized. The correction curve is shown in Figure 5.

### 3.2. Correction of the Micro Humidity Sensor

The constant temperature and humidity testing machine was used as the environmental criteria for the correction of a micro humidity sensor. At different temperatures, the resistivities at the same humidity were different; as such, the temperature was fixed at 25 °C, 50 °C, and 75 °C for correction in the RH range from 40% to 100%. One recording point was made whenever the RH increased by 10%. As there were different resistivities at the same humidity in the processes of humidification and dehumidification, low-humidity-to-high-humidity and high-humidity-to-low-humidity corrections were performed, respectively. After 30 minutes of stabilization, each time, the resistivity of the micro humidity sensor was captured instantly by the NI PXI data acquisition unit; additionally, the correction curve was obtained and nondimensionalized, as shown in Figure 6.

### 3.3. Correction of the Micro Flow Sensor

LEADFLUID BT100S-1 acid and an alkali-resistant, speed-adjusting peristaltic pump were used to provide a steady flow for flow correction. The peristaltic pump conveys fluid by alternately compressing and loosening the rubber tube. The fluid will not touch the mechanical components due to the design of the peristaltic pump; additionally, the fluid will only touch the rubber tube—and, therefore, the mechanical components of the motor will not contaminate the product—thereby preventing unnecessary chemical contamination and guaranteeing the integrity of delivery. The measurable flow range of the peristaltic pump was 30~1700 mL/min, the baseline of the flow correction range was 30 mL/min, and the flow was measured at an interval of 10 mL/min up to total of 100 mL. Three micro flow sensors were corrected, in turn, and nondimensionalized. The correction curve is shown in Figure 7.

### 3.4. Correction of the Micro Pressure Sensor

The Druck-DPI 530 pressure controller (Figure 8), in which the maximum pressure is 20 bar (300 psi), was used to apply fixed pressure to the micro pressure sensor. Meanwhile, the capacitance data were captured by using a Wayne Kerr Electronics 4230 LCR meter (Bognor Regis, UK); the measurable capacitance range of the instrument was 0.01 pF~1 F, and the accuracy was ±0.1%. The micro pressure sensor was corrected under 0~3 bar by fixing the temperature at 20 °C, 30 °C, and 40 °C. It was observed that the capacitance decreased as the temperature rose. As the dielectric layer was of PI polymer, the pressure should be applied repeatedly at the initial stage of correction. The capacitance value was stabilized after unloading, and the dimensionless correction curve of the micro pressure sensor was, lastly, obtained as shown in Figure 9.

### 3.5. Correction of Micro Hydrogen Sensor

The micro hydrogen sensor was corrected using the hydrogen and oxygen supplied from the eight-channel fuel cell testing machine. The micro hydrogen sensor was installed on the runner plate of the high-pressure PEMWE, and the runner was then used as a closed environment for testing. Firstly, the micro hydrogen sensor was connected to an NI PXI data acquisition unit to test the resistance variation; a constant temperature and oxygen at a constant flow rate were set, such that the oxygen ions were adsorbed on the surface of a micro hydrogen sensor. Secondly, a constant temperature and the hydrogen, at a constant flow rate, were set. The presence of hydrogen could lead to the removal of oxygen ions from the surface of the micro hydrogen sensor and a reduction in the sensor resistivity. Thus, the hydrogen was tested by using the resistivity difference of different gases. Figure 10 shows the dimensionless correction of two micro hydrogen sensors from 20 °C to 70 °C. In addition, changes in the resistivity of the micro hydrogen sensor at 25 °C, when the oxygen and hydrogen were supplied, were tested. The results show that the flexible seven-in-one microsensor fabricated in this study produced a response in this environment. The resistivity decreased at the moment when the hydrogen was supplied, and the resistivity increased when the oxygen was supplied, as shown in Figure 11.

## 4. High-Pressure PEMWE

A high-pressure PEMWE was developed in this study, including its structure, the selection of material, and the flow-field design. A columnar runner was used for the flow-field design, as shown in Figure 12. The gas can be carried away rapidly, such that the oxygen can also be discharged rapidly, and thus the water electrolysis reaction area will not be reduced by the gas in the runner. In addition, with the collector plate that was also designed in this study, the high-pressure PEMWE and flexible seven-in-one microsensor were assembled in accordance with the assembly program of related techniques, of which a stereogram is shown in Figure 13.

### 4.1. Real-Time Microscopic Monitoring of High-Pressure PEMWE 

A high-pressure-resistant, flexible seven-in-one microsensor was embedded in the upstream, midstream, and downstream of high-pressure PEMWE in order to monitor the internal state. A 100-hour test was performed to observe the differences among upstream, midstream, and downstream pressures. 

### 4.2. High-Pressure PEMWE Test Environment 

First, the temperature, flow velocity, and the voltage of high-pressure PEMWE must be set. The normal temperature of 25 °C was the optimal operating temperature for a high-pressure PEMWE, and the flow could influence the flow field and temperature distribution inside the PEMWE. This study used a 1.8 V constant voltage for testing, and then the high-pressure PEMWE was operated at a temperature of 25 °C and a flow velocity of 80 mL/min. Second, the high-pressure-resistant flexible seven-in-one microsensor and high-accuracy capture equipment NI PXI cabinet was used for internal and local real-time microscopic monitoring of the high-pressure PEMWE. 

### 4.3. Voltage Test for High-Pressure PEMWE 

The temperature of the DI water in the high-pressure PEMWE was set at 25 °C, the flow velocity at 80 ml/min, and the 100-hour measurement was performed with a constant voltage of 1.8 V. The voltage distributions in the upstream, midstream, and downstream are shown in Figure 14. It can be observed that the change at the upstream inlet was relatively drastic. 

### 4.4. Current Test for High-Pressure PEMWE 

In the environment where the temperature was 25 °C and the flow velocity was 80 ml/min, the 100-hour measurement was performed with a constant voltage of 1.8 V, and the signals were captured every 30 mins. The current distributions in the upstream, midstream, and downstream are shown in Figure 15. It was observed that the current clearly changed at the upstream inlet in the reaction process. 

### 4.5. Temperature Test for High-Pressure PEMWE

The data show that the temperature changes when an upstream current is applied were relatively large in the reaction process, and the temperature changes when the midstream and downstream currents were applied were relatively mild, but also relatively higher. This is because the water in the high-pressure PEMWE generated some heat after the reaction; further, when the temperature rose, which was conducted by water, the temperature at the downstream outlet rose slightly, as shown in Figure 16. 

### 4.6. Humidity Test for High-Pressure PEMWE 

The data show that the upstream, midstream, and downstream relative humidity were 100% in the reaction process, and the resistive micro humidity sensor displayed errors under the temperature effect, as shown in Figure 17. 

### 4.7. Flow Test for High-Pressure PEMWE

Figure 18 shows the 100-hour flow distributions in the upstream, midstream, and downstream flow of the high-pressure PEMWE. It was observed that the upstream of the runner had the highest flow velocity, and the downstream had the lowest flow velocity. This is because the fluid in the columnar runner was relatively smooth and stable in comparison to that in the snakelike runner. 

### 4.8. Pressure Test for High-Pressure PEMWE 

Figure 19 shows the 100-hour upstream pressure test for the high-pressure PEMWE. The pressure was fixed at 3 bar for testing; if there is a leak inside the high-pressure PEMWE, the internal pressure changes noticeably. This experimental result proves that the design and assembly of the high-pressure PEMWE are good, as there were no leaks. 

### 4.9. Hydrogen Test for High-Pressure PEMWE 

Figure 20 shows the 100-hour cathode hydrogen runner outlet test for the high-pressure PEMWE. The micro hydrogen sensor could not touch the oxygen, and it was restored to the state before measurement; therefore, the resistivity decreased noticeably when the hydrogen was introduced.

## 5. Conclusions

In this study, a flexible seven-in-one microsensor, which is resistant to a high-pressure environment, was successfully developed using MEMS technology. The micro voltage, current, temperature, humidity, flow, pressure, and micro hydrogen sensors were successfully integrated onto a 20 μm thick PI film substrate, and the PI (Fujifilm Durimide? PI 9320)—which is resistant to electrochemical corrosion—was used as a protection layer. This high-pressure, resistant, flexible seven-in-one microsensor is characterized by seven simultaneous sensing functions: corrosion resistance, small area, high sensitivity, good temperature tolerance, real-time measurement, and arbitrary placement. 

Three flexible seven-in-one microsensors were successfully embedded in the upstream, midstream, and downstream of the anode runner plate of a high-pressure PEMWE. This was achieved without influencing its operation, and when the temperature, humidity, flow, pressure, and hydrogen that it was subjected to were corrected. The internal and local voltage, current, temperature, humidity, flow, pressure, and hydrogen data of the high-pressure PEMWE were successfully captured by an NI PXI data acquisition unit during the 100-hour operation process of the high-pressure PEMWE. Notably, there was also no damage in the process.

## Figures and Tables

**Figure 1 membranes-12-00919-f001:**
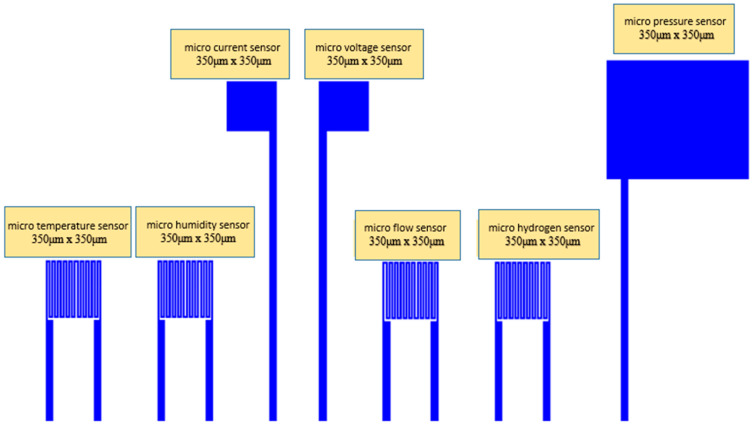
Design drawing of a flexible seven-in-one microsensor.

**Figure 2 membranes-12-00919-f002:**
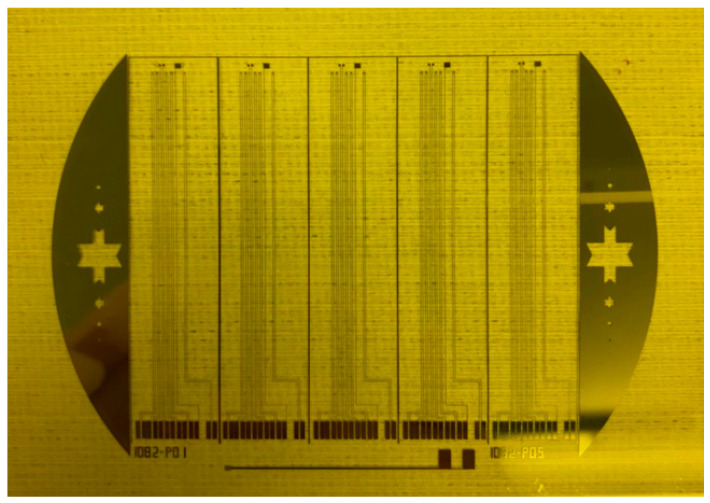
Mask layout of a flexible seven-in-one microsensor.

**Figure 3 membranes-12-00919-f003:**
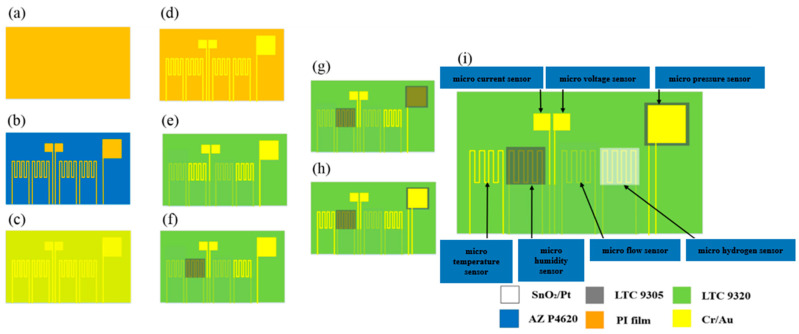
Process flow chart of a flexible seven-in-one microsensor.

**Figure 4 membranes-12-00919-f004:**
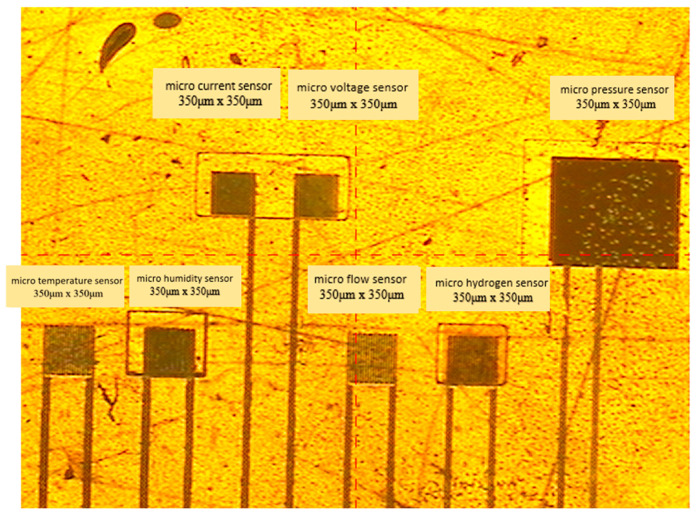
Optical micrograph of a flexible seven-in-one microsensor.

**Figure 5 membranes-12-00919-f005:**
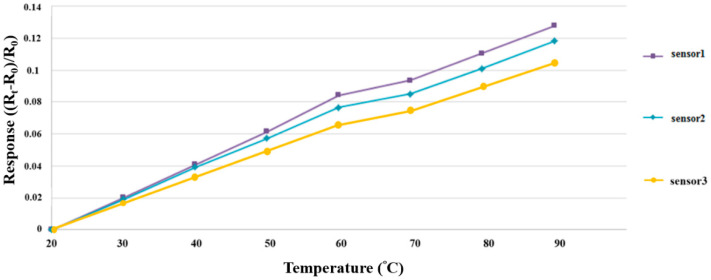
Correction curve of a micro temperature sensor.

**Figure 6 membranes-12-00919-f006:**
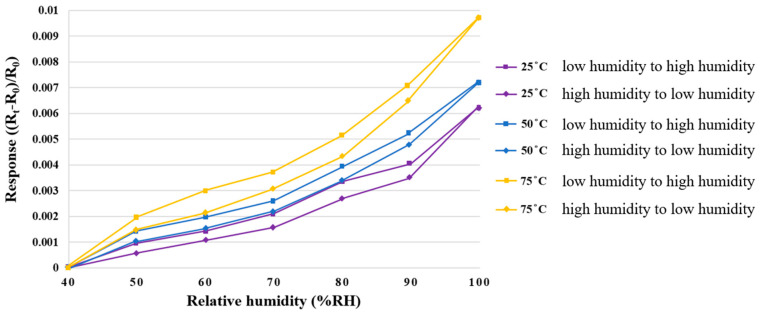
Correction curve of a micro humidity sensor.

**Figure 7 membranes-12-00919-f007:**
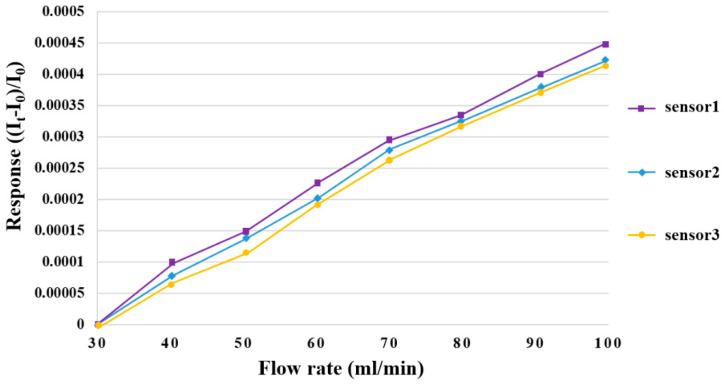
Correction curve of a micro flow sensor.

**Figure 8 membranes-12-00919-f008:**
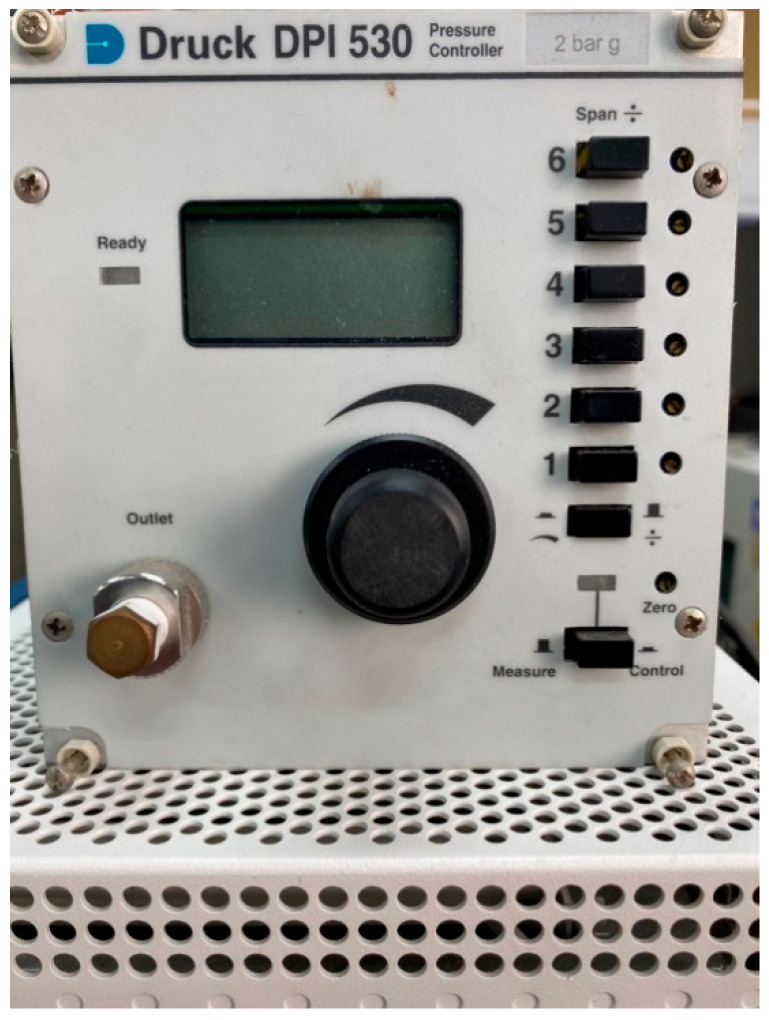
A Druck-DPI 530 pressure controller.

**Figure 9 membranes-12-00919-f009:**
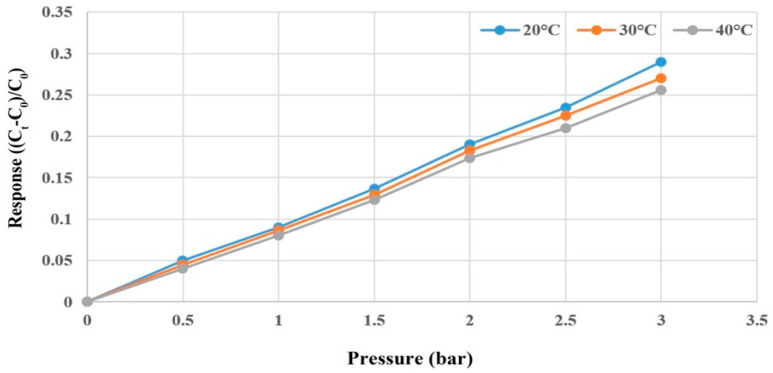
Correction curve of a micro pressure sensor.

**Figure 10 membranes-12-00919-f010:**
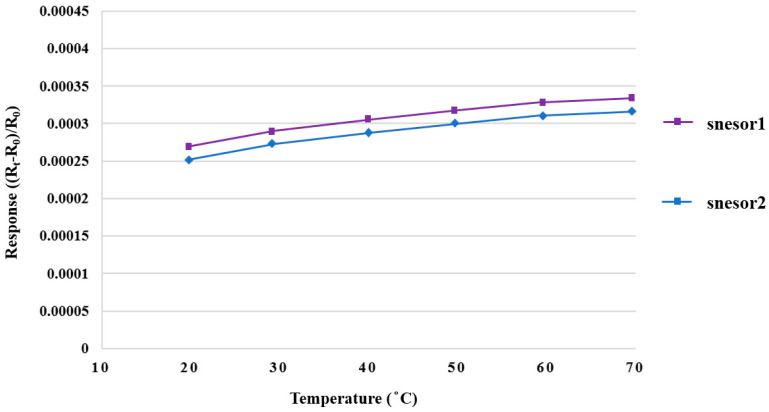
Correction curve of a micro hydrogen sensor.

**Figure 11 membranes-12-00919-f011:**
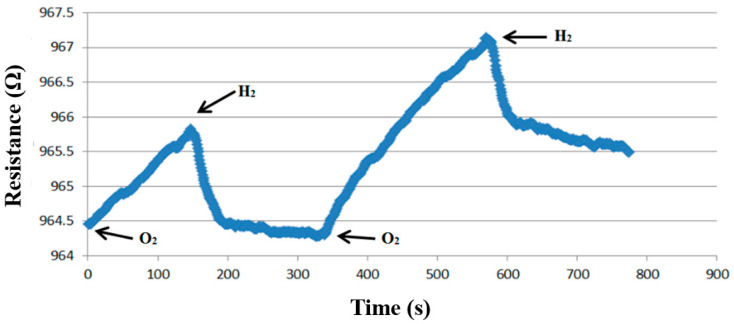
Variation in resistivity of a micro hydrogen sensor at 25 °C when oxygen and hydrogen are supplied.

**Figure 12 membranes-12-00919-f012:**
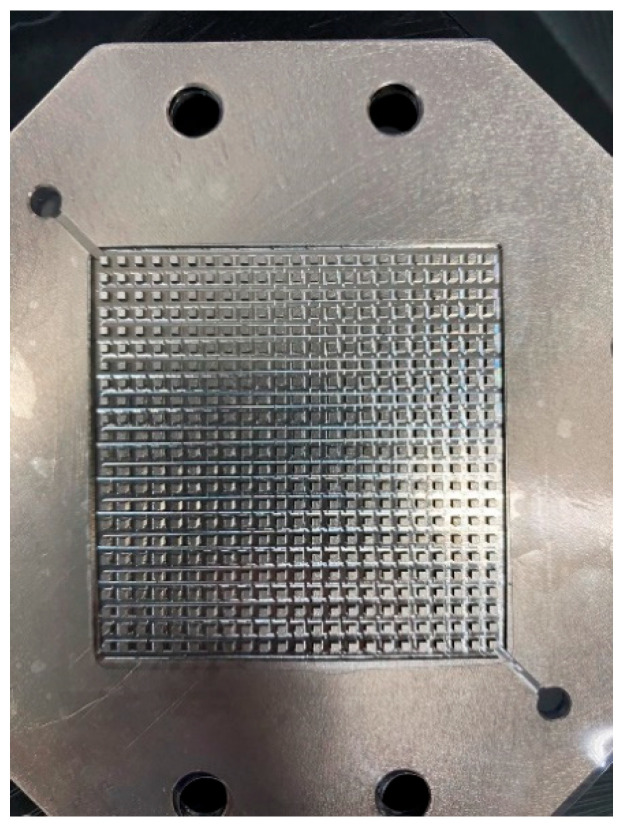
Stereogram of columnar runner inside a high-pressure PEMWE.

**Figure 13 membranes-12-00919-f013:**
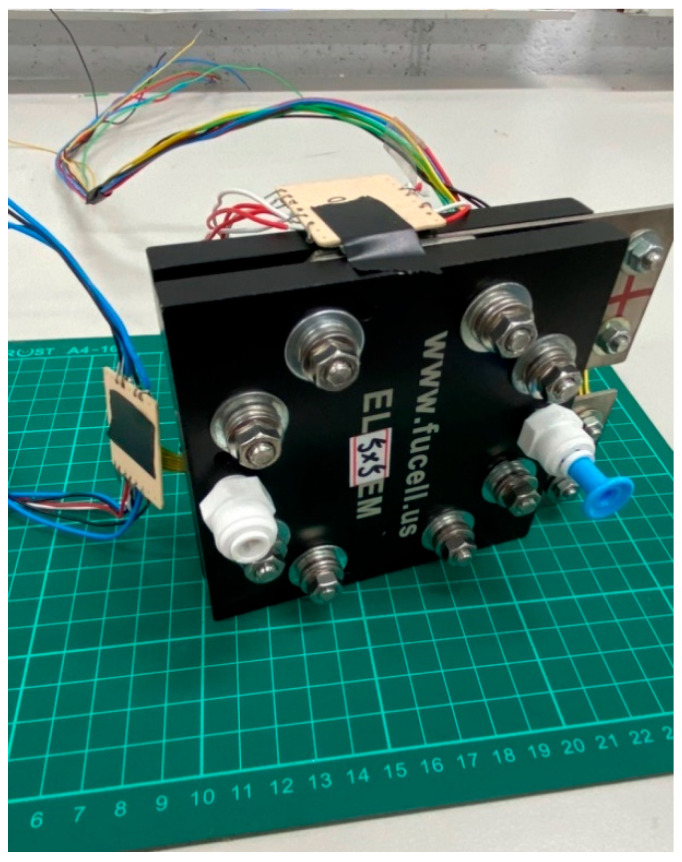
Stereogram of flexible seven-in-one microsensor embedded in a high-pressure PEMWE.

**Figure 14 membranes-12-00919-f014:**
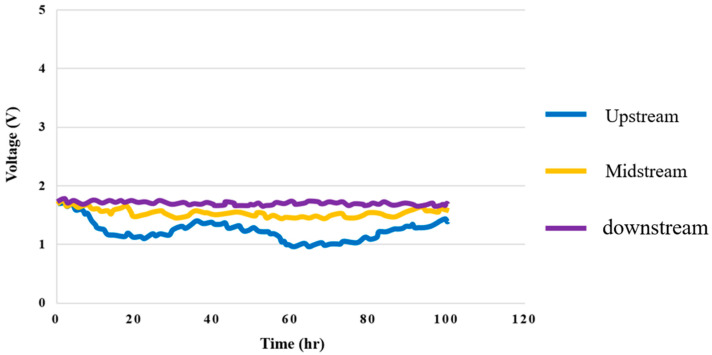
Upstream, midstream, and downstream 100-hour voltage distributions.

**Figure 15 membranes-12-00919-f015:**
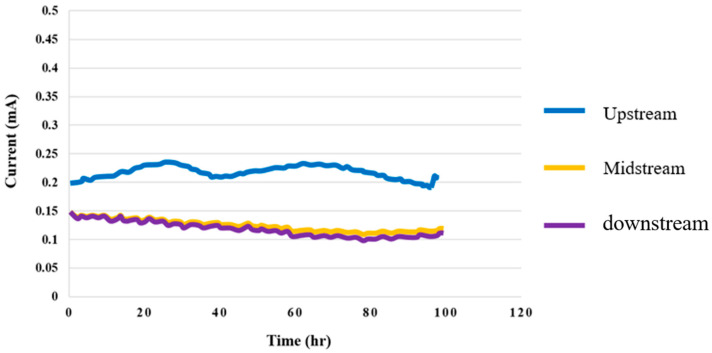
Upstream, midstream, and downstream 100-hour current distributions.

**Figure 16 membranes-12-00919-f016:**
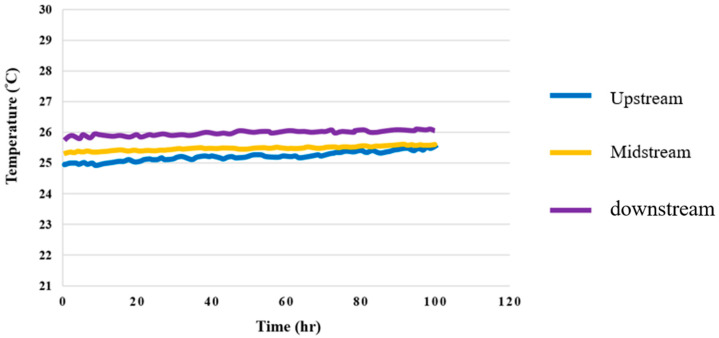
Upstream, midstream, and downstream 100-hour temperature distributions.

**Figure 17 membranes-12-00919-f017:**
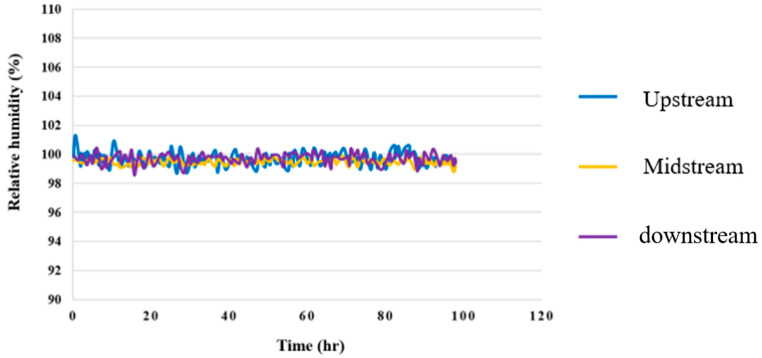
Upstream, midstream, and downstream 100-hour RH distributions.

**Figure 18 membranes-12-00919-f018:**
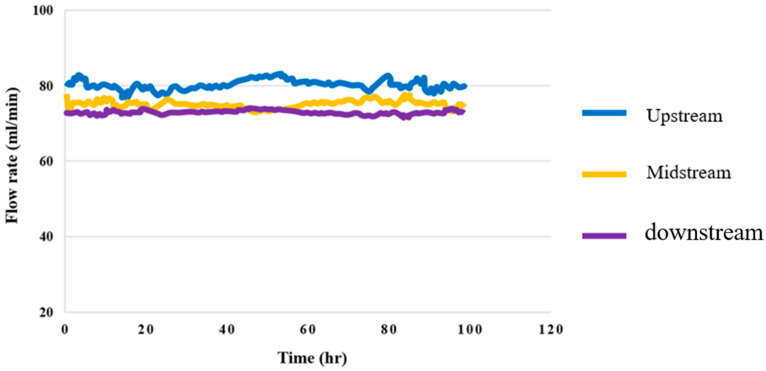
Upstream, midstream, and downstream 100-hour flow distributions.

**Figure 19 membranes-12-00919-f019:**
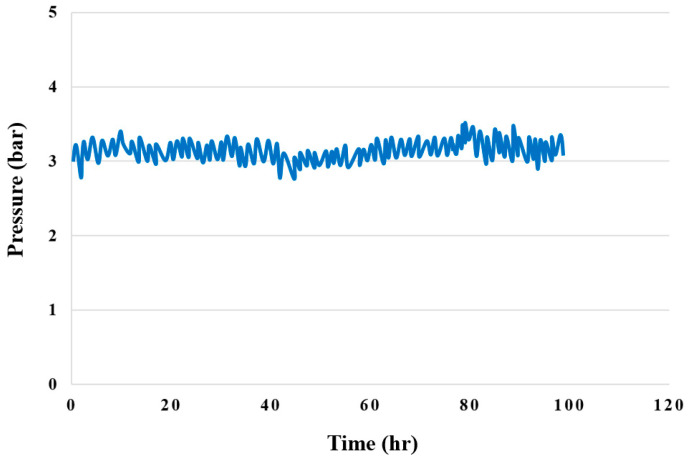
Upstream 100-hour pressure distribution.

**Figure 20 membranes-12-00919-f020:**
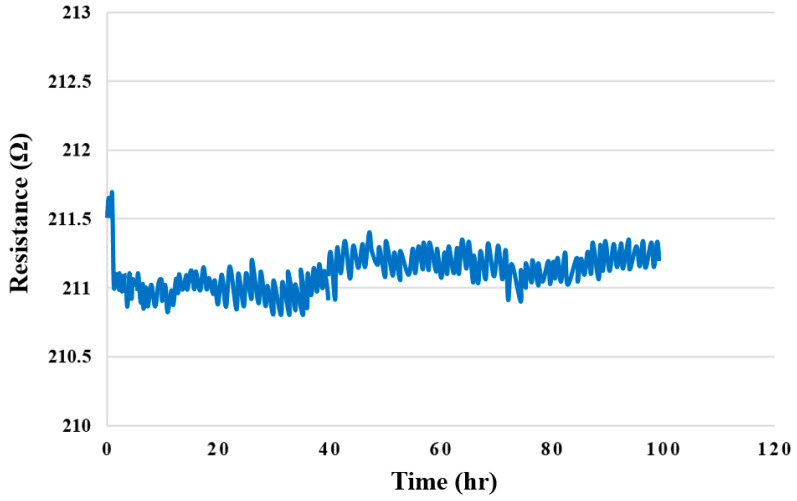
Hydrogen outlet 100-hour hydrogen distribution.

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
