# Peer review of "High-Pressure-Resistant Flexible Seven-in-One Microsensor Embedded in High-Pressure Proton Exchange Membrane Water Electrolyzer for Real-Time Microscopic Measurement"

_membranes, 2022, doi:10.3390/membranes12100919_

Round 1
Reviewer 1 Report
The manuscript reported the innovatively developed high-pressure flexible seven-in-one micro-sensors for the high-pressure PEMWE to realize monitoring in real-time. Multiple parameters can be observed for their effects on hydrogen production. The lift-off process makes the quality of the microsensor more stable. The selection and integration of high-pressure materials for microsensors and PEMWE were also the research and development goals of this paper
The content of this manuscript will definitely meet the reading interests of the readers of the journal. However, the discussion and explanation should be further improved. I suggest giving a minor revision and the authors need to clarify some issues or supply some more experimental data to enrich the content.
1. The abstract is obviously too long, which is far beyond the requirement of the journal. ‘The abstract should be a total of about 200 words maximum.’ [https://www.mdpi.com/journal/membranes/instructions] Hence, the content of the abstract should be further reduced and refined.
2. For grammar issues, it is suggested that the author double-check the small grammar errors in the full text, especially the lack of and redundant use of definite articles.
3. For the Keywords, ‘hydrogen production’, ‘real-time monitor’, ‘aging’, and ‘high pressure’ should be added in order to attract a broader readership.
4. Line 48, ‘It is universally believed that the stability of network systems will be damaged when the coverage of intermittent renewable energy exceeds 20% [4], so the development of renewable energy is the primary issue of various countries' governments.’ The logic of this sentence is questionable. If renewable energy exceeds 20% will damage the stability of the network, why does the government still prefer to develop renewable energy? It should be explained that to reduce the reliance on fossil fuels, governments have to put focus on renewable energy (10.1016/j.electacta.2019.03.056; 10.1016/j.jpowsour.2020.229445). But due to the intermittency, to solve the peak issue and realize peak shaving becomes the primary issue.
‘However, there is a severe defect in the present renewable energy, i.e. seasonal influence.’ Even daily, solar energy peaks and valleys during energy generation.
5. Line 53, ‘Electrical energy storage technology is the key factor in whether renewable energy can reach an economic scale [5, 6].’ While Line 62 suddenly jumps to ‘Shin et al. [10] found that the hydrogen generation of PEMWE was proportional to the supplied current, so the hydrogen production per unit area of the cell stack was proportional to the operable cur- rent density.’
PEMWE is not an energy storage system as a secondary battery. It should be clarified that the combination of PEMWE and PEMFC, can act as an energy storage system.
6. Line 83, ‘The simulation and test results of Liu et al. [16] proved that after the PEMWE had better contact pressure, it would have better electrochemical properties, and the PEMWE with a pneumatic clamping mechanism had a better distribution of the contact pressure.’ What does it mean ‘better contact pressure’? It is better to use higher or lower to be more accurate. Is it due to higher contact pressure increasing the contact and reducing the contact resistance, that leads to better electrochemical performance?
7. Line 182, ‘so the temperature was fixed at 25°C, 50°C, and 75°C for correction in the RH range of 40% to 100%.’ The reason for the selection of such temperatures and RH value ranges should be explained.
8. Line 234, ‘The resistivity decreased at the moment when the hydrogen was supplied, and the resistivity in- creased when the oxygen was supplied, as shown in Figure 11.’ Although this is the measured and observed result, what is the mechanism and background behind this result? What leads to the variations of resistivity?
9. Part 4.2, 4.3, and 4.4, why only constant temperature and flow rate are adopted during the measurements, not a series of values or a range? The current results seem to be monotonous and not comprehensive enough.
10. What are the PEMs used in the PEMWE? This is also very important point that should be clarified.
11. Some more references about PEMWE and sensor should be added, for example, 10.1016/j.ijhydene.2021.01.077, https://doi.org/10.3390/mi12050494, and 10.3390/membranes11020092 .
Reviewer 2 Report
The draft titled " High-Pressure Resistant Flexible Seven-in-One Micro-sensor Embedded in High-Pressure Proton Exchange Membrane Water Electrolyzer for Real-Time Microscopic Measurement" describes the development of a new type of micro hydrogen sensor and integrated flexible seven-in-one micro-sensors (voltage, current, temperature, humidity, flow, pressure, and hydrogen) that are embedded in different positions inside the high-pressure PEMWE while monitoring in real-time. The subject matter and method are interesting to a broad spectrum of readers, but the subject matter does not seem to nicely fit with the journal. As a reviewer, I have noticed some minor corrections that the authors might consider making in their manuscript.
1. The discussion part is very poor and less communicative than the previous reports. The authors should make it much easier to understand by comparing it step by step to similar reports from the past.
2. The future direction part of this manuscript is also very short, which should be glorified.
3. In section 3, correction of the flexible micro-sensor seven-in-one, there are several sub-sections that have discrepancies in data presentation, e.g., a few showing three sets of data, a few two, and again a few ones. Why? Data presentation should be consistent.
4. Throughout the manuscript, the author has not explained most of the curves’ behavior in the figures. Which should I explain properly?
5. Although the authors have used three sets of data for temperature, flow rate, and two hydrogen sensors, they have not mentioned what does represent sensors 1, 2, and 3.
6. All the figure cations should change in a proper manner with all the operating conditions, especially Figures 5, 6, 7, and 9,10.
7. There are lots of typos and chemical symbols in the wrong form. The authors should correct it.
Round 2
Reviewer 2 Report
Even though there have been significant improvements to the work, there are still a few problems that I, as a reviewer, urge the authors to address. These comments should be taken into account as the authors work to improve the paper.
